# Abundance of arthropods as food for meadow bird chicks in response to short- and long-term soil wetting in Dutch dairy grasslands

Livia De Felici[1], Theunis Piersma[1,2] and Ruth A. Howison[1]

[1] Conservation Ecology Group, Groningen Institute for Evolutionary Life Sciences, University of Groningen, Groningen, The Netherlands
[2] Department of Coastal Systems and Utrecht University, NIOZ Royal Netherlands Institute for Sea Research, Texel, The Netherlands

## ABSTRACT

**Background:** Throughout the world, intensive dairy farming has resulted in grasslands almost devoid of arthropods and birds. Meadow birds appear to be especially vulnerable during the chick-rearing period. So far, studies have focused mainly on describing population declines, but solutions to effectively stop these trends on the short-term are lacking. In this study at a single farm, we experimentally manipulated soil moisture through occasional irrigation, to mitigate against early season drainage and create favorable conditions for the emergence of above-ground arthropods during the meadow bird chick rearing phase.
**Methods:** To guarantee the presence of at least a sizeable arthropod community for the measurement of effects of wetting, we selected a farm with low intensity management. The land use and intensity of the study site and surroundings were categorized according to the national land use database and quantified using remote sensing imagery. From May 1 to June 18, 2017, we compared a control situation, with no water added, to two wetting treatments, a “short-term” (3 weeks) treatment based on wetting on warm days with a sprinkler system and a “long-term” treatment next to a water pond with a consistently raised water table from 2010. We measured soil temperature, soil moisture and resistance as well as the biomass of arthropods at 3-day intervals. Flying arthropods were sampled by sticky traps and crawling arthropods by pitfall traps. Individual arthropods were identified to Order and their length recorded, to assess their relevance to meadow bird chicks.
**Results:** The land use analysis confirmed that the selected dairy farm had very low intensity management. This was different from most of the surrounding area (20 km radius), characterized by (very) high intensity land use. The experiments showed that irrigation contributed to cooler soils during midday, and that his happened already in the early part of the season; the differences with the control increased with time. In the short- and long-term treatments, soil moisture increased and soil resistance decreased from the mid-measurement period onward. Compared with the control, cumulative arthropod biomass was higher in the long-term treatment, but showed no change in the irrigation treatment. We conclude that small-scale interventions, such as occasional irrigation, favorably affected local soil properties.

Corresponding author
Livia De Felici,
liviadefelici@gmail.com

However, the effects on above-ground arthropod abundance currently appear limited or overridden by negative landscape-scale processes on arthropods.

## INTRODUCTION

Post-war agricultural intensification of agriculture has negatively altered the ecology of rural landscapes (*Newton, 2004*, *2017*). Herb-rich meadows were replaced by monocultures, foot drains (shallow surface drainage ditches) by underground drainage pipes, while increased grazing pressure and the heavy use of machinery led to the degradation of soil structure and natural soil renewal processes; this resulted in hard dry top soils with low fertility and biodiversity (*Roach & Campbell, 1983*; *EASAC Secretariat, 2018*). These changes were correlated with ongoing declines of arthropods. In German nature reserves, a reduction of 75% was observed between 1989 and 2017 (*Hallmann et al., 2017*), probably following a longer trajectory of decline (*Benton et al., 2002*; *Potts et al., 2010*). Vegetation homogeneity has implicated the loss of habitat for many arthropod species, while the excessive use of agrochemicals contributed substantially to the disappearance of pollinators and other insects as well (*Biesmeijer, 2006*; *Goulson et al., 2015*; *Nilsson, Franzén & Jönsson, 2008*; *Ollerton et al., 2015*; *Vickery et al., 2001*).

Arthropods are integral to healthy terrestrial ecosystem functioning (*Seastedt & Crossley, 1984*; *Yang & Gratton, 2014*). Pollinators are responsible for the sexual reproduction of the majority of flowering plants (*Ollerton, Winfree & Tarrant, 2011*). About one-third of global food production comes from crops that are partially or totally dependent on animal pollination (*Klein et al., 2007*). In the soil, arthropods play fundamental roles in the decomposition processes (*Mattson & Addy, 1975*; *Chapman et al., 2003*; *Panizzi & Parra, 2012*), influencing nutrient cycles directly and indirectly (*Chapman et al., 2003*; *Hawlena & Schmitz, 2010*). Last, but not least, arthropods represent the food source for all those animals who have an insectivorous diet, including about 60% of bird species (*Morse, 1971*).

In the last 40 years, declining European farmland bird populations parallel the arthropod population crash, since meadow birds fail to fledge chicks in environments with low densities of invertebrates (*Kentie et al., 2018*; *Loonstra, Verhoeven & Piersma, 2018*, *2019*; *Schekkerman & Beintema, 2007*). In The Netherlands, the population of one such European endemic farmland bird, the Black-tailed Godwit (*Limosa limosa limosa*), has declined by >70%, with an alarming rate of 6% per year in recent years (*BirdLife International, 2004*; *Kentie et al., 2016*; *Van Dijk et al., 2010*). Similar declines are shown by Eurasian Oystercatcher (*Haematopus ostralegus*), Northern Lapwing (*Vanellus vanellus*), Eurasian Curlew (*Numenius arquata*) and Common Redshank (*Tringa totanus*) (*PECBMS, 2017*; *Van Dijk et al., 2010*). As the diet of small chicks is entirely comprised of above-ground arthropods (*Loonstra, Verhoeven & Piersma, 2018*; *Schekkerman &*

Beintema, 2007), the loss of arthropods negatively affects meadow birds particularly during the breeding season. Food availability is generally hampered directly by the degraded conditions of the soil, with hard and dry top layers that limit the ability of the birds to probe into the ground (Gilroy et al., 2008; McCracken & Tallowin, 2004; Onrust et al., 2019).

Reducing the management intensity of the agricultural fields can improve soil conditions and habitat quality on the long term. Limited input of agrochemicals facilitates the recovery of pollinator species (Frampton, Van Den Brink & Gould, 2000; Goulson et al., 2015), while low additions of organic fertilizer and moderate levels of grazing can encourage sward heterogeneity and benefit invertebrate prey (Vickery et al., 2001). However, these changes require changes in policies on a large scale and require a long time to be implemented. The rapid declining rates in arthropods and birds call for immediate and innovative solutions (see Fuentes-Montemayor, Goulson & Park, 2011).

Soil temperature and moisture are two important factors that influence arthropods presence. Laboratory experiments show that below an optimal range of moisture, the mortality of many arthropods increases (Cho, Rhee & Lee, 2000). Field studies in forests and in agricultural environments also recorded a negative effect of drought on various taxa of soil fauna, with Collembola, Diptera and other predatory arthropods declining under conditions of dry soil and high temperatures (Frampton, Van Den Brink & Gould, 2000; Pflug & Wolters, 2001; Lindberg, Engtsson & Persson, 2002; Tsiafouli et al., 2005). Artificial irrigation can modify soil characteristics and, in some cases, increase the abundance of soil fauna (Frampton, Van Den Brink & Gould, 2000; Lindberg, Engtsson & Persson, 2002). Irrigation is relatively easy for land managers to implement at the individual field scale which, when applied to multiple fields at once, may form the basis for a large-scale management intervention towards improving soil conditions for arthropods and concomitantly meadow birds. However, little is known about the efficacy of short-term irrigation in increasing above-ground arthropods during the breeding season.

In this study, we manipulated wetting conditions in Dutch dairy grassland. To maximize the possibility of encountering a healthy arthropod community, and therefore maximizing the chance for a positive experimental effect, we chose a conventional agricultural dairy farm with low intensity management (Onrust & Piersma, 2017). To verify the actual ecological quality of the habitat, we quantified the land use intensity of the farm and its surrounding using remote sensing data (Howison et al., 2018) and information about land use available in the Dutch national database (Ministerie van Economische Zaken en Klimaat, 2018). The farm already adopted measures to promote habitat for breeding meadow birds, including the construction of a water pond in one of the fields. Therefore, we measured soil conditions (temperature, moisture and resistance) and arthropod biomass under different treatments: a stable high water table in the field that was next to the pond (long-term treatment), and periodical irrigation (short-term treatment) and non-irrigation (control) in an adjacent field. We expected wet and soft soil to offer the best condition for above-ground arthropod community. Therefore, we considered the field with the high water table as the one with the best habitat quality and

predicted improvements in soil conditions and arthropod biomass in the irrigated treatment.

## MATERIALS AND METHODS

### Study site

The experimental study took place at the dairy farm in Wommels, province of Friesland, The Netherlands (53°5′35″N, 5°33′51″E) (Fig. S1). Authorization to work on this area was granted by the land owner, Murk Nijdam and the Cooperative Verening Sùdewestkust. Land use on this farm has been classified as permanent agricultural grassland since at least 2009 (*Ministerie van Economische Zaken en Klimaat, 2018*) and was managed for the protection of breeding meadow birds within the Dutch Agri-Environmental Schemes. The management of the grasslands includes one fertilization per year with farmyard manure: a mixture of straw, cattle dung and urine collected and composted for up to a year (*Onrust & Piersma, 2017*). Mowing of all meadows takes place after June 15, because it is assumed that the majority of meadow bird chicks has fledged after this date. The mowing is followed by a period of grazing that continues until October or November. Water is drained by an underground system of pipes, while foot-drains are absent. All the grasslands of the farm have clay soils. Between May and June temperatures usually range from a minimum of 9–12 °C to a maximum of 16–18 °C and the average precipitation is 17.2 mm (*Koninklijk Nederlands Meteorologisch Instituut (KNMI), 2018*). The construction in one of the fields of a water pond of approximately 90 × 50 m began in 2009 and finished in 2010.

### Contextualization of the landscape

In order to determine the ecological quality of the surrounding landscape, we analyzed the spatial footprint of different land use intensities at increasing buffer distances from the study farm (2, 5, 10 and 20 km with the proposed study site as the central point). Land use was categorized with the Dutch national land use database (*Ministerie van Economische Zaken en Klimaat, 2018*) over four buffer zone distances. Land use intensity, referring to the amount of disturbance, was quantified using the variation surface roughness measured by the Sentinel-1 C-SAR (active radar) satellite and verified with detailed ground surveys (see *Howison et al., 2018* for a detailed description).

### Wetting experiment

Two herb-rich meadows of respectively, 2.8 and 5.4 ha were chosen. In the first grassland an irrigation pipe with six sprinklers was installed. The pipe crossed the land diagonally, from the northwest to the southeast corner (Fig. S1). The sprinklers were placed within 50 m from each other and had a reach of 12 m. The pipe was connected to a pump that drained water from an adjacent canal. The system was manually activated when the farmer expected a warm day and it was on for a minimum of 5 min to a maximum of 70 min (Fig. S2). The short-term meadow was divided into four blocks, two irrigated (short-term) and two non-irrigated (control). Each block contained two replicates for the measurements of soil temperature and arthropod abundance, placed 15 m from each other. The second grassland, with the water pond, was located 100 m south-west from the

meadow with the short-term experiments. A set of two replicates was placed in this grassland at equivalent distances (~35 m) from the pond and the field margin.

## Vegetation and soil parameters

One 50 m transect was laid out perpendicular to the irrigation pipe to account for both the irrigation treatment effect closest to the pipe (distance 0–12 m), and the control treatment beyond the reach of the irrigation pipe (distance 20–50 m). In the field near water (long-term treatment) a 50 m transect was laid out 25 m from the edges of the field to avoid any edge effects and orientated in the same direction as the field with control and irrigation treatment. Vegetation height (±1 cm) was measured at one m intervals along the transects by lowering a one m vertical measuring rod into the vegetation to the soil surface and drawing the 10 closest leaves their full vertical height. Plant species touching the rod at each one m interval were identified (*Streeter et al., 2009*).

Soil temperature was measured by Thermochron® iButton® devices (DS1921G) located at each replicate, sealed into small plastic bags and attached to the surface of the soil. The loggers were programed to record the temperature every hour starting from the 0.00 on May 1, 2017 until the end of the experiment at 0.00 on June 19, 2017.

Soil moisture was measured at one m intervals along the transects using a ML3 Theta probe (ML3-UM-1.0; EijkelkampAgrisearch Equipment), with settings: device = ML2 and soil type = organic. To account for the full range of well-drained to water-logged soils, field capacity was set to 0.999 $m^{-3}$.

Soil penetration was measured at one m intervals along the transects using a hand-penetrometer for top-layers (Type IB, EijkelkampAgrisearch Equipment). The internal springs used were 100N, Ø 1.6 mm for soft moist soils and 150N, Ø 1.75 mm for dry hard soil. The force used to push a 0.25 $cm^2$ cone to a depth of 11 cm into the soil (the depth important both for emerging arthropods and probing meadow birds (*Lourenço et al., 2010*) was calculated as: Resistance ($N/cm^2$) = (Total force (cm) × Spring force (N/cm))/Cone diameter ($cm^2$), thereafter converted with a constant factor to $kg/cm^2$. Soil moisture, soil penetration pressure, vegetation composition and height were surveyed at three moments during the season, i.e., early (May 1), midterm (May 17) and late (June 8, 2017) (Fig. S2).

## Arthropods

Arthropods were sampled over intervals of 3 days between May 1 and June 18, after this date the meadows were mowed. The experiment was stopped as the mowing disrupts any season-long monitoring of arthropod biomass and confounds the eventual effects of the wetting experiment. Sticky traps were used to collect flying arthropods. The traps consisted of yellow plastic boards of 10 by 60 cm coated in a thin layer of non-drying glue (Bug Scan®; Biobest Group NV, Westerlo, Belgium). In each replicate, the sticky boards were positioned facing a north-south orientation. All the arthropods on the traps were identified to Order and their lengths measured to the nearest mm. Pitfall traps were used to collect crawling arthropods. They consisted of transparent plastic containers (300 ml) buried into the ground with the rim on the surface. The containers were half filled with a

mixture of ethylene glycol and water (1:4) and were refilled approximately once a week to prevent complete evaporation or excessive dilution in case of rain. Arthropod biomasses were calculated using the length-weight equations from *Rogers, Buschbom & Watson (1977)*.

### Data analysis

Variation in soil temperatures during the day were analyzed using a generalized additive model (GAM) with a normal distribution from the R package mgcv (*Wood, 2011*). The dataset was divided into two periods: early season (May 1–May 16, 2017) and late season (May 17–June 8, 2017) and analyzed separately for days with and without irrigation events. Temperature was used as a response variable, while the treatments and date were used as predictors. Differences in soil moisture, soil resistance and vegetation height among the treatments were investigated using one-way ANOVA, post hoc group contrasts were analyzed using Tukey's HSD from the R package Agricolae (*Mendiburu, 2013*) with 95% confidence intervals. Soil moisture was regressed against soil resistance using a linear exponential model.

The yields of sticky traps and pitfall traps were analyzed separately. Treatment effects on the variation in arthropod biomass during the season were analyzed for each Order separately, considering only Orders that represented at least 1% of the cumulative biomass. GAMs with $\gamma$ distribution were used to analyze arthropods biomass (the accumulation of arthropods over 3-day intervals), with date and treatment as predictor variables (*Zuur et al., 2009*). To account for the difference in sample size between treatments, cumulative biomass was calculated for the duration of the experiment; differences between treatments were compared using effect size ratios (*Hedges, Gurevitch & Curtis, 1999*). To analyze the composition in size, the biomass was divided into three length classes: big (≥4 mm), small (two to three mm) and very small arthropods (one mm). Land use intensity, categorized into different land use types (*Ministerie van Economische Zaken en Klimaat, 2018*), was analyzed with one-way ANOVA for each buffer distance, and post hoc Tukey HSD was used to determine significantly different groups (*Mendiburu, 2013*). All analyses were performed using R 3.3.1 (*R Core Team, 2017*).

## RESULTS

### Contextualization of the landscape

Land use categories showed variation in land use intensity (represented by variation in C-SAR1: Fig. 1), with the lowest intensity use in protected areas and semi-natural grasslands, followed by agricultural grasslands and temporary grasslands and highest intensity use in the arable land. Land use intensity of the categories differed significantly in each buffer zone: two km (ANOVA: $F_{(3,473)} = 23.4$, $R^2 = 0.12$, $P < 0.001$), five km (ANOVA: $F_{(3,2368)} = 13$, $R^2 = 0.01$, $P < 0.001$), 10 km (ANOVA: $F_{(4,8897)} = 269.5$, $R^2 = 0.10$, $P < 0.001$), and 20 km buffer zones (ANOVA: $F_{(4,21953)} = 1,301$, $R^2 = 0.19$, $P < 0.001$) (Fig. 1A). The land use intensity of the study farm, characterized as agricultural grassland (Fig. 1A), but scored lower than that of the protected areas (Fig. 1B).

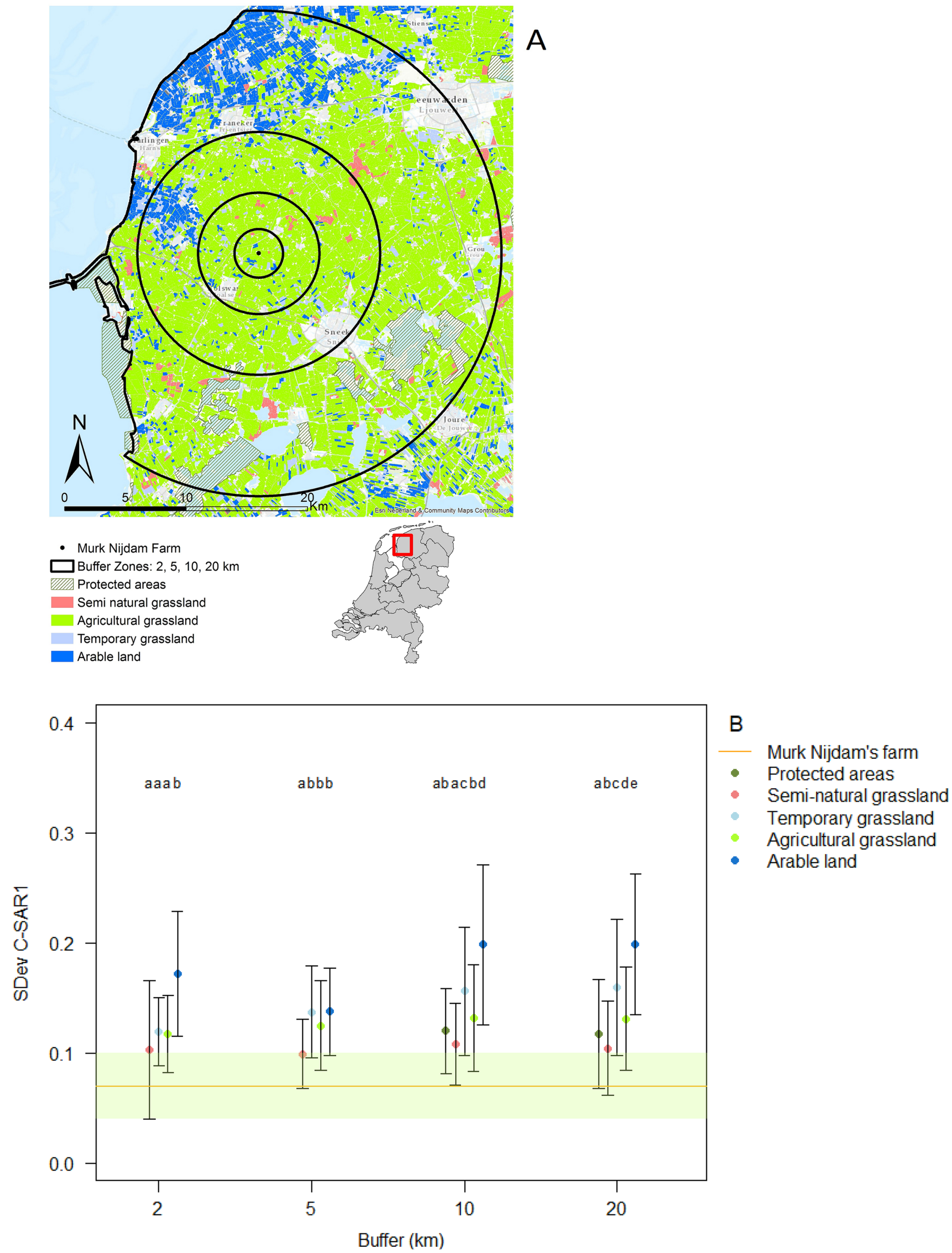

**Figure 1 Classification of land use and land use intensity variation.** Image created using data from Basisregistratie Gewaspercelen (BRP). (A) Agricultural land use was divided into buffer distances of 2, 5, 10 and 20 km surrounding the study farm and categorized as different land use types; (B) variation in land use intensity was classified into different land use types (color codes in A and B are identical). The horizontal line represents the average land use intensity of our study site ± SD. Different letters represent significant differences $P < 0.05$ (Tukey HSD), within buffer distances.

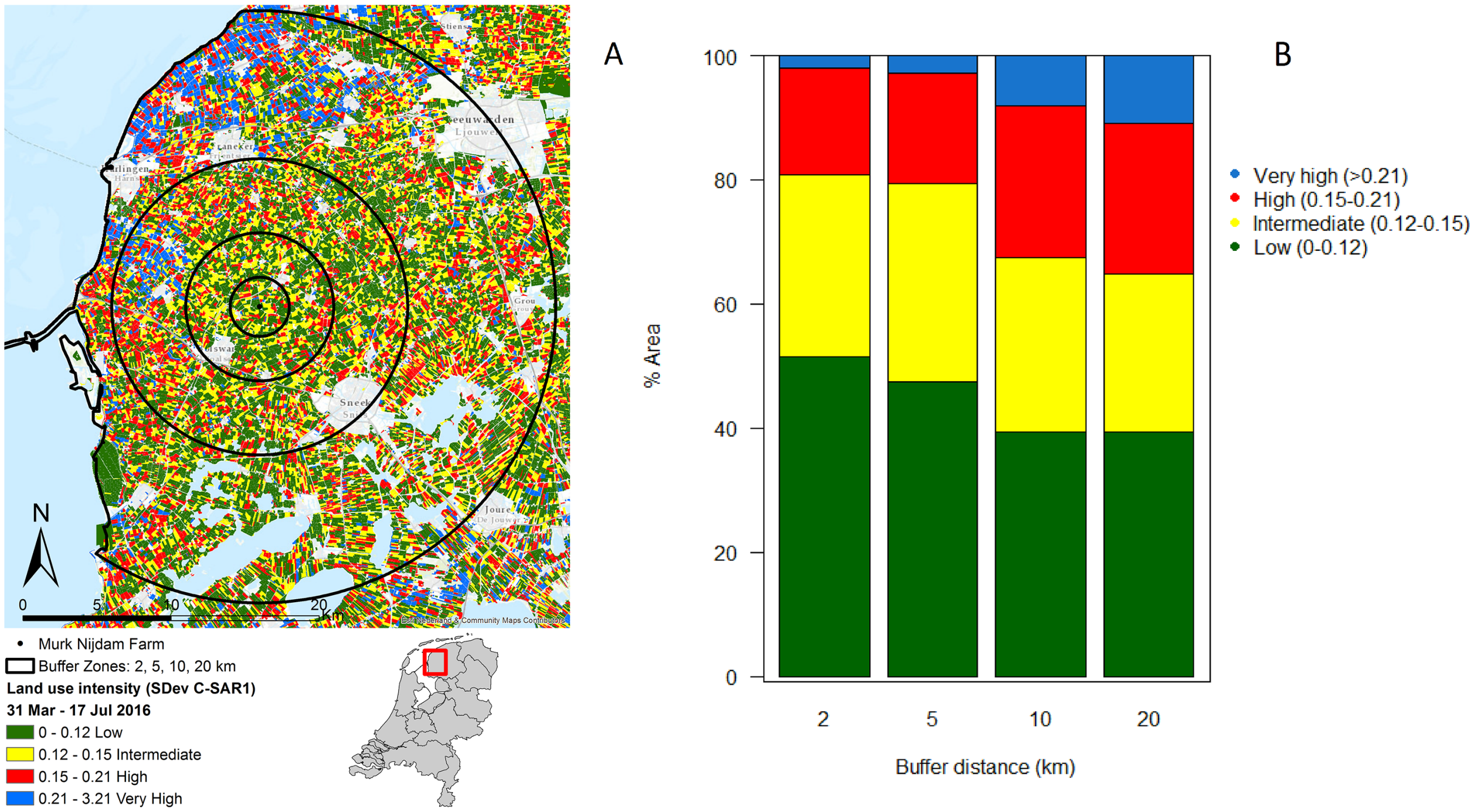

**Figure 2 Agricultural land use intensity.** Agricultural land use intensity divided into buffer distances of 2, 5, 10 and 20 km surrounding the study farm, summarized as (A) the standard deviation of change in Radar derived surface roughness (SDev C-SAR1); (B) proportion of land under different land use intensities within agricultural fields surrounding the study farm. Image produced with the data from ESA remote sensing data, sentinel 1 CSAR and processed by Ruth A. Howison, University of Groningen.

In the immediate proximity (two km buffer) of the study farm 50% of the land is under intermediate or high intensity management, which increases to 60% within a radius of 20 km from the farm (Figs. 2A and 2B).

## Soil parameters

Between 13:00 and 15:00 h, soil temperatures daily reached peaks in all treatments (Figs. 3A and 3B). During the days with irrigation ($N = 4$), in the early season the highest values were reached in the control treatment (26.1 ± 0.8 °C) (Fig. 3A), while in the short-term and long-term treatments the maxima were lower (short-term: 24.3 ± 0.7 °C; long-term: 19.5 ± 0.7 °C). The GAM model revealed a significantly different pattern of variation between treatments, especially between the control and the long-term treatments. On dry days without irrigation events ($N = 18$), the highest temperature was reached again in the control treatment (25.5 ± 0.5 °C), followed by short-term (24.0 ± 0.4 °C), and the long-term treatment (19.8 ± 0.4 °C). In this case, the GAM revealed different patterns of variation, either for the control and the long-term treatment, than for the control and short-term treatment (Table 1). During the late season, temperatures were higher (Fig. 3B). On days with irrigation events ($N = 5$), the highest values were registered in the control treatment (31.3 ± 0.5 °C), followed by the short-term treatment (25.1 ± 0.4 °C) and the long-term one (20.6 ± 0.3 °C). The variation in temperature over time was significantly different between treatments (Table 1). Similarly, during days without irrigation ($N = 7$) the highest peaks were in the control treatment (32.0 ± 0.5 °C), followed

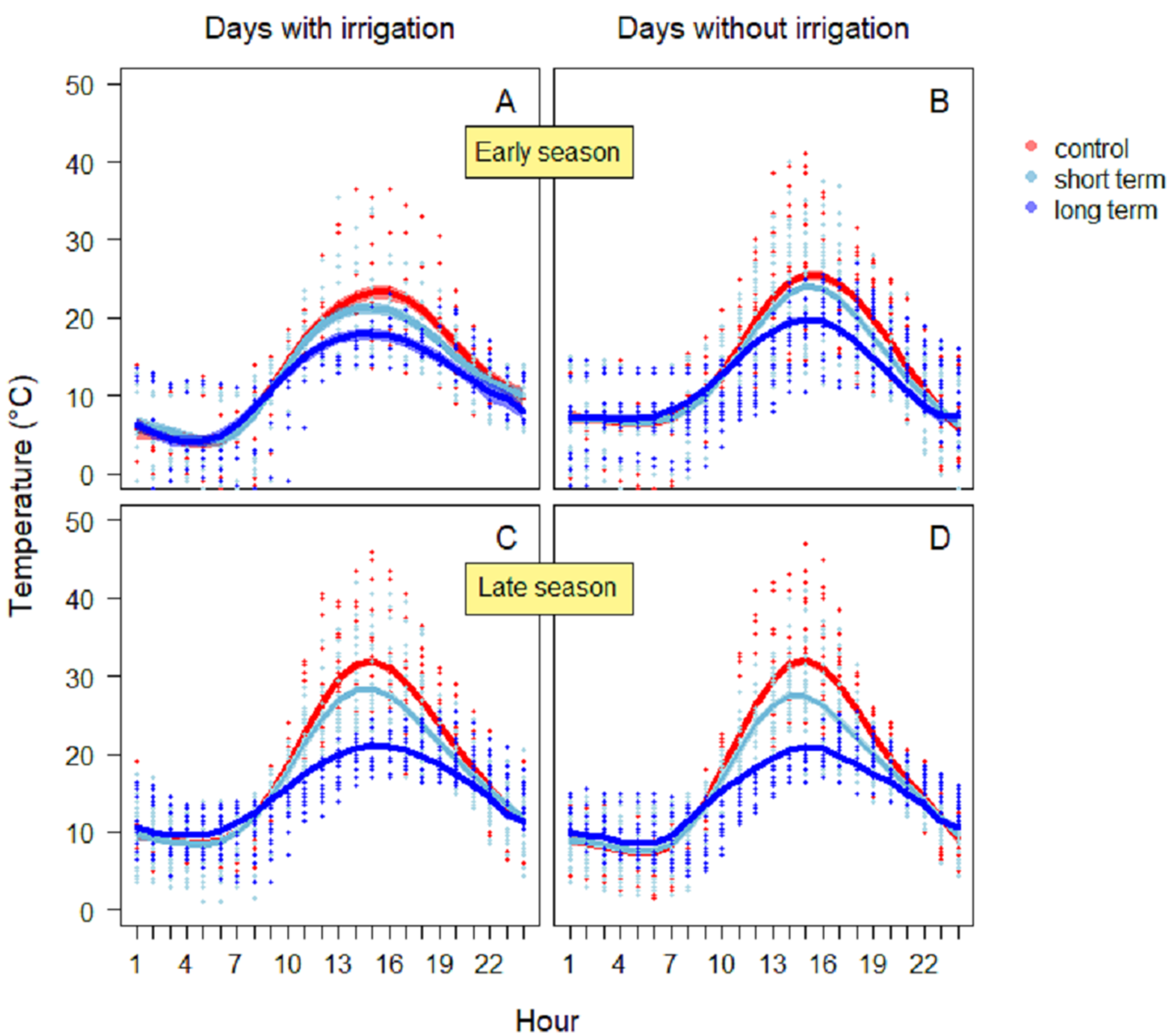

**Figure 3 Soil temperature variation during the early season in days with irrigation (A) and without irrigation (B) and during late season with irrigation (C) and without irrigation (D).** The solid red line follows the smoothed trend for the control (without added water) treatment, dark blue for the long-term (high water table) treatment and light blue the short-term (irrigation) treatment, the shaded area in the respective color represents ± SD.

**Table 1 Generalized additive model fit of soil temperature to treatment using time (hours) as smoothing term.**

| Temperature season | Days with irrigation | Treatment | | s (h) | | | $R^2$ | Deviance explained (%) |
|---|---|---|---|---|---|---|---|---|
| | | *F*-value | *P*-value | *F*-value | Edf | *P*-value | | |
| Early | Yes | $F_{2,684}$ = 8.54 | <0.001 | 129.3 | 7.01 | <0.001 | 0.62 | 62.7 |
| | No | $F_{2,1728}$ = 19.21 | <0.001 | 366.2 | 7.14 | <0.001 | 0.64 | 64 |
| Late | Yes | $F_{2,1512}$ = 59.45 | <0.001 | 457.9 | 7.04 | <0.001 | 0.72 | 72 |
| | No | $F_{2,1512}$ = 59.76 | <0.001 | 497.7 | 7.64 | <0.001 | 0.74 | 74.5 |

**Note:**
Edf refers to the effective degrees of freedom for the smoothing spline.

by the irrigation (27.5 ± 0.4 °C) and then the long-term treatment (21.0 ± 0.3 °C). In this case as well, the variation in temperature over time was significantly different between treatments (Table 1).

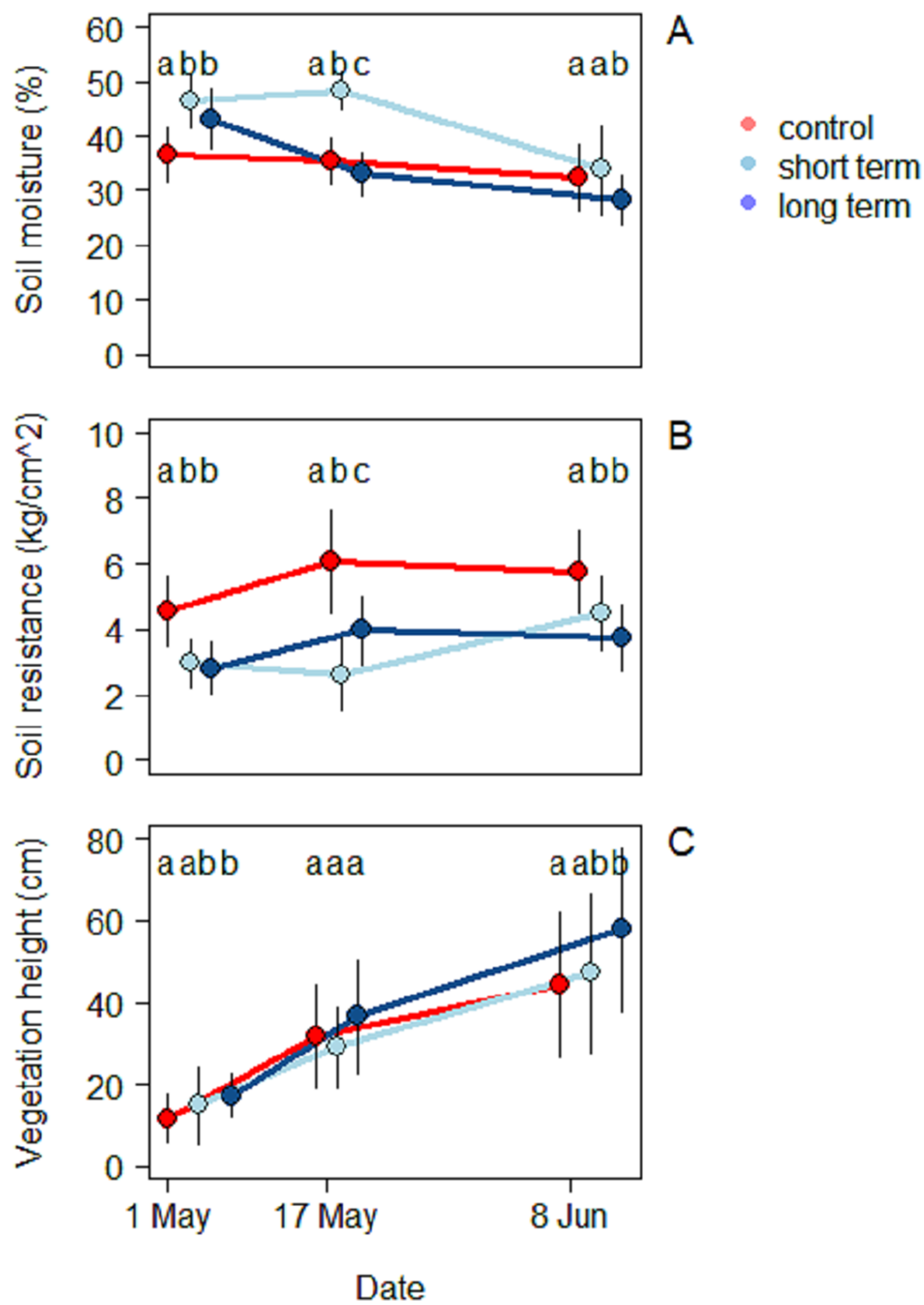

**Figure 4 Soil characteristics and vegetation at the beginning (May 1), middle (May 17) and end of the season (June 8), under different treatments namely; control (without added water), long-term (high water table) and short-term (irrigation).** (A) Soil moisture (%), (B) soil resistance (kg/$cm^2$) and (C) vegetation height (cm).

In all three treatments, soil moisture decreased during the season (Fig. 4A). During the first survey on May 1 the highest moisture level was recorded in the short-term treatment (46.5% ± 4.9%), followed by the long-term one (43.2% ± 5.6%), while the control treatment was significantly dryer (36.7% ± 4.92%) (ANOVA: $F_{(2,90)} = 20.6$, $R^2 = 0.31$, $P < 0.001$). During the second survey, the short-term site kept the highest moisture values (48.3% ± 5.5%), followed this time by the control (35.4% ± 3.5%) and finally by the long-term treatment (33.1% ± 4.0%). The mean moisture value was significantly different for each treatment (ANOVA: $F_{(2,90)} = 70.8$, $R^2 = 0.61$, $P < 0.001$). At the third and last survey on June 8 the short-term treatment was still the one with the highest moisture value (33.8% ± 8.2%), but there was no significant difference with the control anymore

(28.3% ± 4.4%). The long-term treatment (28.3% ± 4.4%), on the other hand, had a significantly lower level of moisture (ANOVA: $F_{(2,90)} = 70.9$, $R^2 = 0.15$, $P < 0.001$).

Soil resistance increased in the course of the season in all three treatments (Fig. 4B) (2.8 ± 0.8 kg/cm$^2$). During the first survey the highest resistance was recorded in the control treatment (4.5 ± 1.1 kg/cm$^2$) and was significantly different from the short-term (2.9 ± 0.7 kg/cm$^2$) and the long-term treatments (2.8 ± 0.7 kg/cm$^2$) (ANOVA: $F_{(2,90)} = 30.7$, $R^2 = 0.46$, $P < 0.001$). During the mid-season survey soil resistance was highest in the control treatment (6.1 ± 1.6 kg/cm$^2$), followed by the long-term (3.9 ± 1.1 kg/cm$^2$) and the short-term treatments (2.6 ± 1.1 kg/cm$^2$) (ANOVA: $F_{(2,90)} = 41.7$, $R^2 = 0.48$, $P < 0.001$). In the third survey, once again the highest soil resistance was observed in the control treatment (5.8 ± 1.3 kg/cm$^2$), followed by the short-term (4.5 ± 1.1 kg/cm$^2$) and long-term treatments (3.7 ± 1.0 kg/cm$^2$) (ANOVA: $F_{(2,90)} = 31.8$, $R^2 = 0.41$, $P < 0.001$). Soil resistance was correlated with the level of moisture by an inverse proportion relationship (Exponential LM: $F_{(1,298)} = 110$, $R^2 = 0.27$, $P < 0.001$).

A total of 23 different plant species were identified. Species richness was similar in the two transects, but the proportion in which the plants were present was different (Table S1). The short-term grassland presented a predominance of herbaceous species (*Taraxacum officinale*, *Trifolium pratensis*, *Rumix acetosa*, *Ranunculus acris*), while graminoid species dominated the grassland near water (long-term treatment) (*Dactylis glomerata*, *Alopercus pratensis*, *Bromus hordaceous*, *Elytrigia repens*, *Poa trivialis*). Vegetation height increased progressively in all three treatments (Fig. 4C). At the beginning of the season, vegetation was highest in the long-term treatment (17.3 ± 5.5 cm), followed by the short-term (14.9 ± 9.4 cm) and the control treatments (11.7 ± 5.9 cm) (ANOVA: $F_{(2,90)} = 7.7$, $R^2 = 0.15$, $P < 0.001$). In this case only the first and the last one differed significantly. During mid-season there were no significant differences among treatments and the highest vegetation was still found in the long-term treatment (36.6 ± 13.9 cm), followed by the control (31.7 ± 12.6 cm) and the short-term ones (29.0 ± 9.8 cm) (ANOVA: $F_{(2,90)} = 2.3$, $R^2 = 0.05$, $P < 0.001$).

By the end of the season the situation in trend was similar to the beginning, with the long-term treatment being the one with highest vegetation (57.8 ± 19.9 cm) followed by the short-term (47.2 ± 19.6 cm) and the control treatments (44.4 ± 17.8 cm). As in the beginning of the season, the only significant difference was found between the long-term treatment and the control one (ANOVA: $F_{(2,90)} = 5.1$, $R^2 = 0.10$, $P < 0.001$).

## Arthropod biomass

The main arthropod Orders present in the sticky traps were Diptera (80.5%), Lepidoptera (12.4%), Coleoptera (2.8%), Hemiptera (2.5%) and Hymenoptera (1.7%). Aranaea, Acari and Collembola were also present, but contributed <1% to the total biomass, therefore were not used in further analysis (Table S2).

For Diptera, the pattern of variation between sticky trap biomass in the long-term treatment and the control treatment differed significantly (Table 2). Arthropod biomass was generally higher in the long-term treatment than the other two treatments. Diptera biomass peaked on May 22 with mean biomass of 464 ± 74 mg. In the control and

**Table 2** Generalized additive models fit of arthropod biomass to treatment using date as smoothing term for sticky traps.

| Sticky traps Order | Treatment | | s (date) | | | $R^2$ | Deviance explained (%) |
|---|---|---|---|---|---|---|---|
| | *F*-value | *P*-value | *F*-value | Edf | *P*-value | | |
| Diptera | $F_{2,160}$ = 10.57 | <0.001 | 29.23 | 8.76 | <0.001 | 0.76 | 56.9 |
| Lepidoptera | $F_{2,160}$ = 0.27 | 0.77 | 1.65 | 0.38 | <0.001 | 0.38 | 56.1 |
| Coleoptera | $F_{2,160}$ = 0.28 | 0.75 | 4.4 | 8.79 | <0.001 | 0.41 | 38.1 |
| Hemiptera | $F_{2,160}$ = 9.81 | <0.001 | 14.92 | 7.66 | <0.001 | 0.81 | 80.3 |
| Hymenoptera | $F_{2,160}$ = 2.65 | <0.1 | 8.09 | 7.49 | <0.001 | 0.42 | 41.2 |

**Note:**
Edf refers to the effective degrees of freedom for the smoothing spline.

short-term treatments the peak was narrower, with the maximum on the same date, but lower biomass immediately before and after (Fig. 5A).

For Hemiptera there was a significant difference in the patterns of sticky trap biomass in the long-term treatment compared to the control and short-term ones (Table 2). The main peak in the last two was reached June 12–15, with a mean biomass of 42 ± 9 mg in the short-term treatment and 37 ± 1 mg in the control treatment. In the long-term treatment, flying Hemiptera were consistently low or absent (Fig. 5D).

For the other Orders (Lepidoptera, Coleoptera and Hymenoptera) there were no significant differences among treatments in the patterns of variation during the season (Table 2). Lepidoptera appeared mainly in the last part of the season, small peaks were visible in the long-term treatment around the June 3 and in the control treatment around June 9 (Fig. 5B). Coleoptera reached the maximum abundance during the last part of the season, with the highest peaks in the control (60 ± 49 mg) and in long-term treatments (32 ± 24 mg) on June 18 (Fig. 5C). Hymenoptera showed a constant, low-abundance pattern during the season, with two shallow peaks in the short-term treatment on the 17 (12 ± 2 mg) and May 31 (8 ± 2 mg) (Fig. 5E). The comparison of treatment effects on cumulative biomass showed no effect of the irrigation treatment and a positive effect of the long-term treatment (Table 3). On average, less than 10% of the individuals from each sampling events had a size of ≥4 mm (8.5%, SD = 4.5), while the vast majority was small, with a length of two or three mm (43.1%, SD = 11.7) or very small, with a length of one mm (48.3%, SD = 11.8) (Fig. 6).

In the pitfall traps, the composition consisted mainly in Coleoptera (40.1%), Aranaea (33.3%), Diptera (12.6%) and Lepidoptera (7.3%). Samples presented also minor quantity (<1%) of Hymenoptera, Coleoptera larvae and Lepidoptera larvae, Hemiptera, Collembola and Acari (Table S2). All the GAM models for the different Orders of the pitfall traps showed the date affecting biomass over the season ($P < 0.001$), but no difference in the pattern of variation among treatments (Table 4). Coleoptera showed a pattern with a progressive increase by the end of the season, with the highest peak in the control field (111 ± 5 mg) (Fig. 7A). Aranaea had a small peak in all the treatments during the first half of the season, between May 19 and 22 (control: 57 ± 1 mg, short-term: 46 ± 9 mg, long-term: 66 ± 3 mg) (Fig. 7B). Diptera were most abundant in the pitfall traps shortly after the

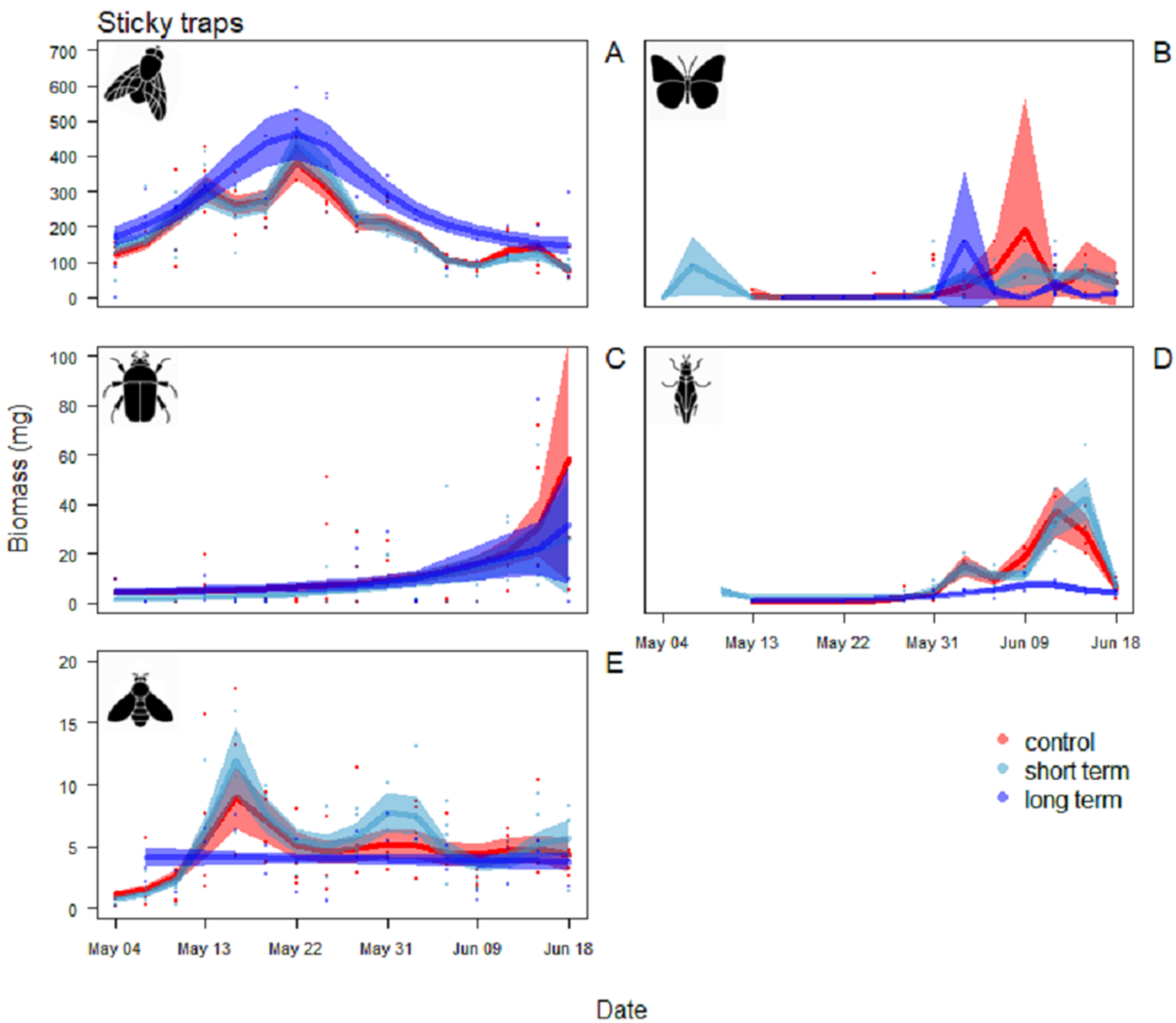

**Figure 5 Biomass variation found in the sticky traps over the season.** (A) Diptera, (B) Lepidoptera, (C) Coleoptera, (D) Hemiptera and (E) Hymenoptera. The graphs are shown in order of decreasing biomass, note the different scales on the *y* axes. The solid red line follows the smoothed trend for the control (without added water) treatment, dark blue for the long-term (high water table) treatment and light blue the short-term (irrigation) treatment, the shaded area in the respective color represents ± SD.

beginning of the season (control: 19 ± 6 mg, short-term: 20 ± 5 mg, long-term 22 ± 8 mg), while showed a constant and low-abundance pattern during the rest of time (Fig. 7C). Lepidoptera showed a constant pattern during the whole season, with no peaks in any particular treatment (Fig. 7D). Compared with the sticky traps, the ratio between cumulative biomass in the short-term and long-term sites against the controls but there was still a small positive effect of the treatments (Table 3). The average of arthropods with a size ≥4 mm in each sampling events was higher than in the sticky traps (31 %, SD = 12). Very small individuals with size of one mm represented almost half of the samples (49%, SD = 14), while arthropods with size two to three mm constituted the rest (20%, SD = 10) (Fig. 8).

## DISCUSSION

The experimental soil wetting was carried out in two fields classified as conventional agricultural grasslands with low land use intensity, in accordance with the intended

**Table 3** **Summary of the log ratio differences comparing the cumulative arthropod biomass of the control treatment (no water added) ($N$ = 4) to the irrigation treatment ($N$ = 4) and near water treatment ($N$ = 2).**

| Replicates | Sticky traps | | Pitfall traps | |
|---|---|---|---|---|
| | Short-term | Long-term | Short-term | Long-term |
| 1 | 0.151 | 0.289 | 0.456 | 0.481 |
| 2 | 0.124 | 0.106 | 0.223 | −0.018 |
| 3 | −0.032 | – | −0.043 | – |
| 4 | −0.059 | – | −0.276 | – |
| Average | 0.046 | 0.197 | 0.090 | 0.232 |

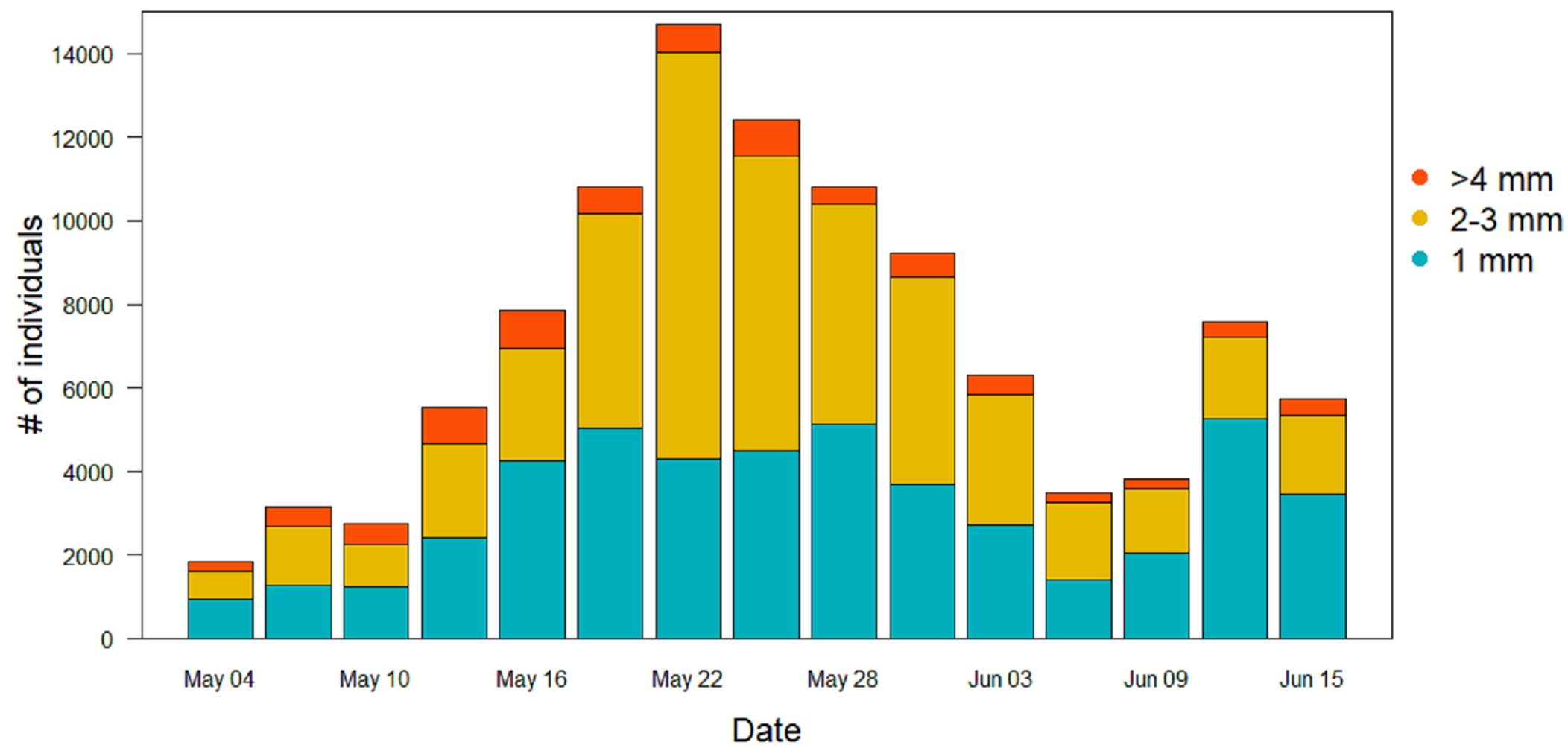

**Figure 6** **Size distribution of the arthropods during the season in the sticky traps.** Individuals with a length ≥4 mm are in orange, two to three mm individuals are in yellow, and one mm individuals are in light blue.

**Table 4** **Generalized additive models fit of arthropod biomass to treatment using date as smoothing term for pitfall traps.**

| Pitfall traps Order | Treatment | | s (date) | | | $R^2$ | Deviance explained (%) |
|---|---|---|---|---|---|---|---|
| | $F$-value | $P$-value | $F$-value | Edf | $P$-value | | |
| Coleoptera | $F_{2,160}$ = 0.15 | 0.86 | 14.55 | 1 | <0.001 | 0.08 | 7.96 |
| Aranaea | $F_{2,160}$ = 0.87 | 0.42 | 5.35 | 3.65 | <0.001 | 0.13 | 21 |
| Diptera | $F_{2,160}$ = 0.38 | 0.68 | 8.54 | 5.50 | <0.001 | 0.35 | 30.5 |
| Lepidoptera | $F_{2,160}$ = 0.20 | 0.82 | 4.88 | 7.70 | <0.001 | 0.45 | 86.6 |

**Note:**
Edf refers to the effective degrees of freedom for the smoothing spline.

provision of meadow bird breeding habitat (*Onrust & Piersma, 2017*). The strengths of this study are its experimental and comparative character, the detailed observations on soils and arthropods and the novel landscape contextualization. The quantification of land use intensity added a new and valuable dimension to the national land use categories,

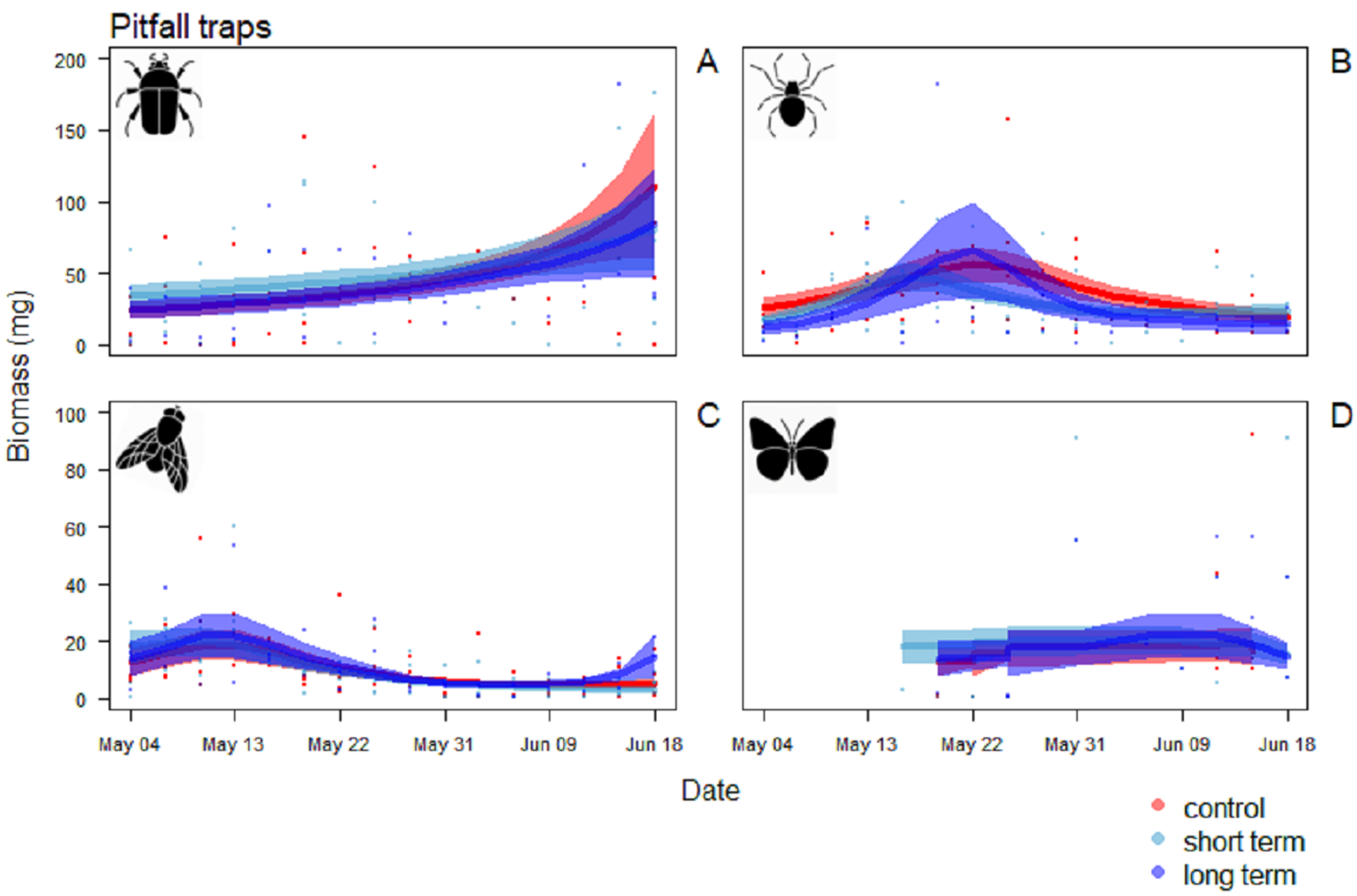

**Figure 7** **Biomass variation in found in the pitfall traps over the season.** (A) Coleoptera, (B) Aranaea, (C) Diptera and (D) Lepidoptera. The solid red line follows the smoothed trend for the control (without added water) treatment, dark blue for the long-term (high water table) treatment and light blue the short-term (irrigation) treatment, the shaded area in the respective color represents ± SD.

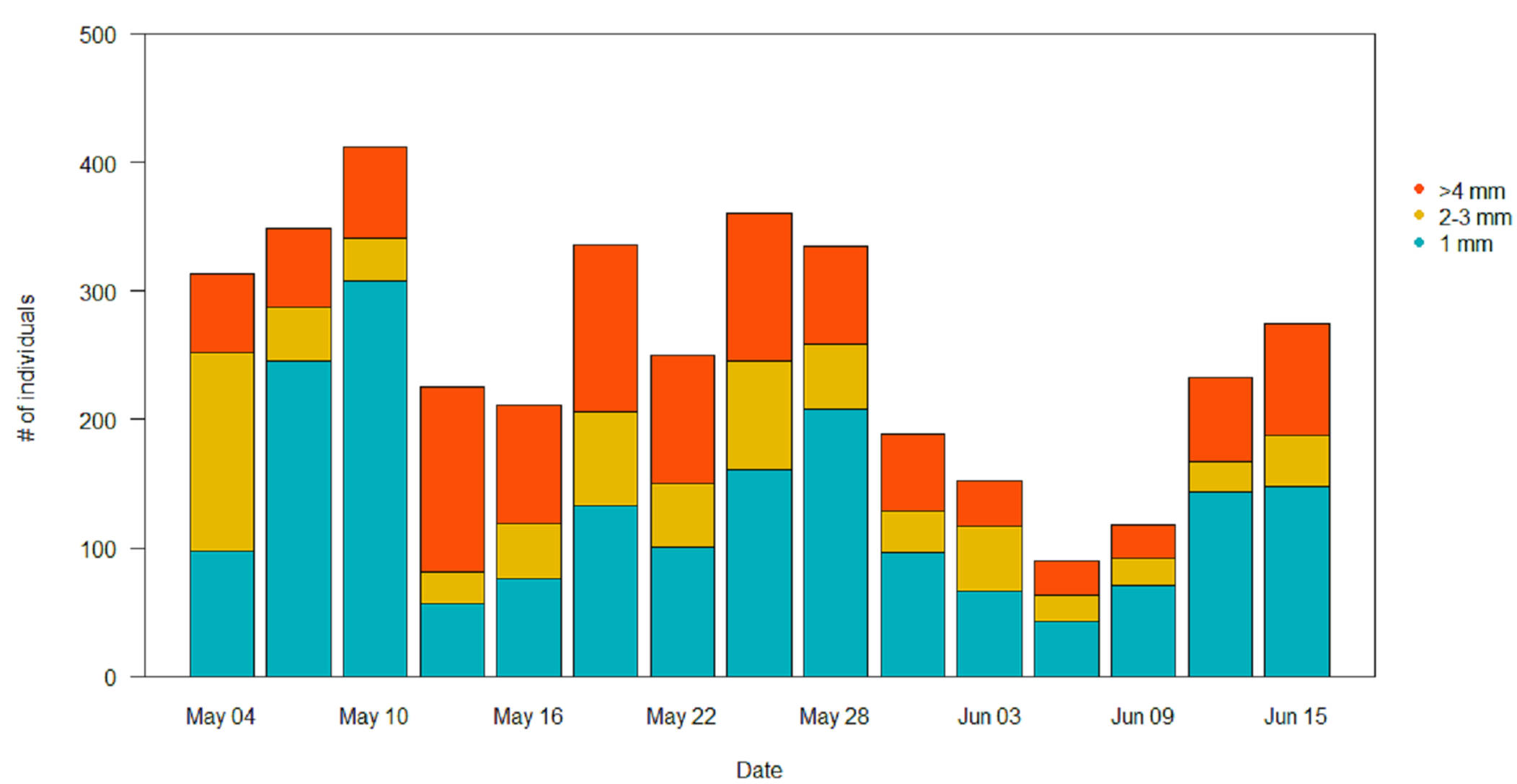

**Figure 8** **Size distribution of the arthropods during the season in the pitfall traps.** Individuals with length ≥4 mm are in orange, individuals of two to three mm are in yellow, and individuals of one mm are in light blue.

spatially contextualizing these low intensity managed fields within a neighborhood of high intensity managed agricultural grasslands. We see this study as a starting point: the fast development of technologies to measure arthropod abundances and diversity with automated image recognition (*Martineau et al., 2017*), should now make it possible to redo this type of study with adequate replication across farms and across time.

Soil temperature, moisture and resistance are key factors in many phases of arthropods life-cycles. The occasional wetting was able to modify characteristics of the soil keeping the soil cool, moist and soft. Irrigation did affect ground temperature already early in the season, with greater differences in temperature peaks as the season proceeded. The addition of water also affected soil moisture, keeping it significantly higher until mid-season, while soil resistance was always lower in the short-term treatment rather than in the control. *Hulthen & Clarke (2006)* showed that for some Diptera extreme moisture conditions have significant influence on pupal survival, with high mortality in dry soils and less likelihood to penetrate them to pupate. Eggs are vulnerable to desiccation and high temperature can accelerate hatching rates but decrease larval survival (*Johnson et al., 2010*). Laboratory studies on different species showed effects on the regulation of the diapauses and the development of the larvae (*Cho, Rhee & Lee, 2000*; *Dimou et al., 2003*; *Ellis et al., 2004*; *Neven, 2000*). Thus, the wet conditions created in this study are expected to favor arthropods emergence (*Hamza & Anderson, 2005*; *Hulthen & Clarke, 2006*; *Johnson et al., 2010*). Nevertheless, the effect on invertebrate biomass in the short-term treatment was small. Since the irrigation only started at the beginning of the breeding season, it is possible that the beneficial effects of the irrigation on arthropods biomass would only have become evident after the sampling period, when the eggs and larvae that were in the soil during the experiment would have hatched and emerged. In any case, such delayed effects of the irrigation on arthropod biomass would not have benefitted meadow bird chicks, growing up during the short window of time covered by our sprinkler treatment (*Kentie et al., 2018*; *Loonstra, Verhoeven & Piersma, 2018*). We note that the vast majority of the arthropods was either small (two to three mm) or very small (one mm). The pitfall traps had higher proportion of bigger individuals (≥4 mm) than the sticky traps, although the cumulative biomass in these traps was remarkably lower. According to *Beintema et al. (1991)*, arthropods of this size would not be large enough to sustain growing meadow bird chicks.

The overall conditions of lower temperatures, elevated initial moisture and soft soil were associated with a higher cumulative arthropod biomass in the long-term treatment. This is consistent with the idea that stable wet conditions promote egg-laying opportunities for arthropods. As soil temperatures were consistently lower than in the control and short-term wetting treatments, the stable high water table would have provided a buffer to temperature fluctuations, keeping the soil cooler throughout the day. These may have been beneficial for the soil fauna, as heat peaks like those recorded in the control and short-term treatments may negatively affect arthropod larvae and adult survival (*Gilbert & Raworth, 1996*; *Neven, 2000*).

Plant communities shape arthropod communities (*Perner et al., 2005*; however, see *Schaffers et al., 2008*), so differences in vegetation might have contributed to the differences

between fields where the experiment was conducted. Differences in plant species composition might also be behind the uneven presence of the Hemiptera, that were completely absent in the long-term treatment. Almost all the individuals sampled from this Order were leafhoppers (Cicadellidae) and the distribution of these plant-sucking insects is tightly related to presence of the host plant species (*Biederman, 2002*). The proximity to a pond in the long-term treatment might have contributed to the abundance of flying arthropods, who came close to the water for feeding or courtship (Fig. 5) (*Drake, 2001*).

The landscape analysis revealed that the management intensity of the dairy farm where the study took place is even lower than in protected areas (Fig. 1B). Therefore, the study farm should yield good chances of finding a healthy invertebrate community. However, the farm is embedded in a landscape with high intensity use. In fact, the analysis revealed that half of the fields in the surroundings of the farm have intermediate or high intensity of usage, and the percentage of high intensity use increases with distance. Furthermore, the data on land use classification indicates that some of the low intensity fields that are present are monocultures and therefore with very limited diversity. Local factors, such as management practices, and regional factors, such as distance to high-diversity habitats, determine local biodiversity (*Tscharntke & Brandl, 2004*). It seems likely that the effects of the wetting were limited or overridden by negative landscape-scale processes.

While current studies meticulously document and call attention to the alarming loss of biodiversity from the 1950s (*Lister & Garcia, 2018*; *Newton, 2017*; *WWF, 2018*), innovative short-term management actions are needed to mitigate against the ongoing trajectory of decline. The wetting experiments showed how important moisture is to improve soil conditions. Nevertheless, the restoration of biodiversity-rich agricultural landscapes requires landscape-wide changes in agriculture.

## CONCLUSIONS

We hypothesized that the occasional irrigation in a dairy farm would improve soil conditions and enhance arthropod emergence during the period of rapid growth by meadow bird chicks. The landscape analysis confirmed that the experimental farm had low levels of land use and a good chance to offer a relatively healthy invertebrate community. We found that the (long-term) treatment of stable high water provided more arthropod biomass to prospective young meadow birds than short-term water irrigated and control fields. Irrigation made soils cooler, moister and softer, but on the short term the arthropod biomass did not visibly respond. Moreover, and perhaps reflecting the wider landscape context, the arthropods sampled were generally too small to be considered suitable food for meadow bird chicks. Thus, we emphasize the urgency of finding innovative solutions to stop biodiversity loss in agricultural environments on the short- and the long-term.

## ACKNOWLEDGEMENTS

We thank farmer Murk Nijdam for his ideas to enable this test of some of his ideas on the availability of arthropods for meadow bird chicks and for facilitating and carrying out

the actual wetting experiment. We are grateful for the encouragement and feedback by Jouke Altenburg and we thank the anonymous reviewers for constructive comments.

### Funding

This work was supported by Vogelbescherming Nederland-BirdLife Netherlands who financed the contribution of Livia De Felici, whilst the 2014 Spinoza Premium of the Netherlands Organization for Scientific Research (NWO) financed Theunis Piersma and Ruth Howison. The funders had no role in study design, data collection and analysis, decision to publish, or preparation of the manuscript.

### Grant Disclosures

The following grant information was disclosed by the authors:
Vogelbescherming Nederland-BirdLife Netherlands.
2014 Spinoza Premium of the Netherlands Organization for Scientific Research (NWO).

### Competing Interests

The authors declare that they have no competing interests.

### Author Contributions

- Livia De Felici performed the experiments, analyzed the data, contributed reagents/materials/analysis tools, prepared figures and/or tables, authored or reviewed drafts of the paper.
- Theunis Piersma conceived and designed the experiments, authored or reviewed drafts of the paper, approved the final draft.
- Ruth A. Howison performed the experiments, analyzed the data, contributed reagents/materials/analysis tools, prepared figures and/or tables, authored or reviewed drafts of the paper.

### Field Study Permissions

The following information was supplied relating to field study approvals (i.e., approving body and any reference numbers):

Authorization to work on this area was granted by the land owner, Murk Nijdam and the Cooperative Verening Sùdewestkust (Farming collective union).

### Data Availability

Data is available in the Supplemental Files and Dataverse:

De Felici, Livia; Piersma, Theunis; Howison, Ruth A., 2019, "Replication Data for: Above-ground arthropod biomass responses to short- and long-term soil wetting in Dutch dairy farmland," https://hdl.handle.net/10411/AXFJSX, DataverseNL, V1.

## Supplemental Information

Supplemental information for this article can be found online at http://dx.doi.org/10.7717/peerj.7401#supplemental-information.

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
