# Peer review of "Abundance of arthropods as food for meadow bird chicks in response to short- and long-term soil wetting in Dutch dairy grasslands"

_PeerJ, doi:10.7717/peerj.7401_

## Round 0.1 · original submission · Major Revisions

Dear Dr. De Felici and colleagues:

Thanks for submitting your manuscript to PeerJ. I have now received three independent reviews of your work, and as you will see, one reviewer recommended rejection, while the other two suggested substantial revisions. I am affording you the option of revising your manuscript according to all three reviews, but understand that your resubmission will likely be sent to at least one new reviewer for a fresh assessment.

The reviewers raised many concerns about the manuscript. Please note that reviewer 1 has provided a marked-up version of your resubmission. All of these concerns need to be addressed, especially those regarding the experimental design. There seems to be a strong lack of clarity when describing both the methods and results. Please address this in your revision.

Aside from the criticisms raised by the reviewers in their reports, be sure to enlist the assistance of an English expert before submitting your revision.

Therefore, I am recommending that you revise your manuscript accordingly, taking into account all of the issues raised by the reviewers. I do believe that your manuscript will be closer to publication form once these issues are addressed.

Good luck with your revision,

-joe

Reviewer 1 ·

Basic reporting

Manuscript entitled “ Above-ground arthropod biomass response to short- and long-term soil wetting in Dutch dairy farmland” provide valuable information and is scientifically correct and well written although there are few grammatical mistakes. Manuscript meets the journal standards. I recommend to accept it after making changes to the manuscript. I have proofread the manuscript using track changes

Experimental design

experiment design is good and results meet the aim of the manuscript.

Validity of the findings

Manuscript fulfils the AIM of the study. I have not seen much work on this topic.

Additional comments

Following corrections should be made prior to the acceptance of the manuscript.

28 should be “very” insted “a very”
75 should be “intensively” instead of “intensive”
77 should be “areas, where farming takes place in ecologically benign forms,” instead “ areas where farming takes place in ecologically benign forms”
98 should be “were” instead “was”
121 should be “the adjacent canal” instead “adjacent canal”
175 should be “the analysis.” instead of “the analysis,.”
172 should be “into two periods” insted “in two periods”
188 should be “divided into three” insted “divided in three”
222 should be “the late season,” insted “the late season”
228 should be “Also, in” insted “Also in”
286 should be “mean,” insted “a mean”
296 should be “irrigated” insted “irrigate”
334 should be “in control” insted “in the control”
349 should be “in control” insted “in the control”
361 should be “insects are” insted “insects is”

Annotated reviews are not available for download in order to protect the identity of reviewers who chose to remain anonymous.

Reviewer 2 ·

Basic reporting

Text is awkward in many places. There are also some spelling and punctuation troubles; proofreading should have caught these errors. To illustrate, a few of the problems, consider these (there are others):
-In the summary, last line of Methods followed by extra “diet.” and first Results sentence is hard to understand.
-L65: “composed of” is used incorrectly the proper verb and construction would be “diet of chicks entirely comprises arthropods”
-L89 needs to be reworked.
-L118, change “On” to “In”, line 121 needs an “an” before adjacent canal.
-L175, random comma at end of sentence.
-Rework L197

-Results summary from first page: “The land use analysis revealed a very low intensity management in the fields of the surrounding (20 km radius) area which was characterized by (very) high intensity land use.” This sentence seems like a contradiction.

Experimental design

The main limitation of the work appears to be the experimental design. If the goal was to test the influence of land-use intensity, I question whether this design was appropriate. It seems appropriate to study a range of land-use intensity. The chosen farm had low levels of land use and thus authors appear to have thought it had the highest chance to harbor good insect populations; however, perhaps the farm has a depauperate insect population for some historic, climatic or some other land use prior to 2009—a low population that the short duration of wetting could not overcome. Perhaps nearby farms have applied some other management strategy or tactic that is greatly limiting insect populations in the region. By only having one farm in one year, authors have no idea. They did not generally test the influence of wetting, they tested the influence of wetting on that farm in that year. A more appropriate experimental design may find different results. An ideal study design would have many focal farms embedded in many landscapes that varied in the complexity of their land use; this has been the common recent approach for testing the influence of landscapes. This paper provides an ambitious design to which author might aspire:
Happe, A.K., Alins, G., Blüthgen, N., Boreux, V., Bosch, J., García, D., Hambäck, P.A., Klein, A.M., Martínez-Sastre, R., Miñarro, M. and Müller, A.K., 2019. Predatory arthropods in apple orchards across Europe: Responses to agricultural management, adjacent habitat, landscape composition and country. Agriculture, Ecosystems & Environment, 273, pp.141-150.

Studying only one farm in a landscape context seems problematic. Low replication, or more likely pseudo-replication (two fields within the same farm, subject to the same landscape), do not appear to have provide a robust design. Even if one does not agree that two plots of each kind within one farm is pseudo-replication, then having just two replicates for one year, is problematic. The design is not robust.

The near-water field should be considered a control, as opposed to a treatment. This field is described as a treatment throughout the manuscript (lines 23, 32, 91), yet no treatment is applied to this field. It is more like a positive control.

Validity of the findings

The most robust result is that adding water made soil moist, cooler, and softer. This is not very notable.

The effect of the irrigation treatment on arthropod abundance was small (L336); this would be expected after just one season of irrigation. Some insect larvae spend multiple years in the soil, and even univoltine populations would likely take more than a year to respond to lower soil temperatures and higher soil moisture. This study would require more field sites (in multiple landscapes with varying management intensities) and multiple study years (as briefly noted in L391) to confidently conclude that landscape effects overshadow in-field irrigation. The research conclusion (L405-406) is a bold statement to make from an experiment conducted at only one location, in a single year, with only two replicates per treatment. A more suitable conclusion would be based on short-term difference (or lack of differences) caused by irrigation.

Invoking a landscape explanation for the lack of an influence of wetting on arthropod abundance seems hard to defend because only one farm in one landscape was studied. Would authors have obtained the same results in a different landscape, say one that was surrounded by forest?

Additional comments

Overall, I appreciate their research goal is solution-focused, as opposed to just pointing out a problem.

Specific comments:
Were ibuttons buried, or on the soil surface? Clarify this in L140.

Line 254-259 discusses the differences in plant composition across sites, yet in line 354-356 this is contradicted. There may be value in performing NMDS with plant species composition.

Also it may be valuable to run NMDS with arthropod communities, especially since there does appear to be a community shift between the near-water site and the irrigated field (dipterans and hemipterans).

Would be valuable to include more discussion of the effects of soil temperature on insect survival and emergence rate (expand upon L327)

Reviewer 3 ·

Basic reporting

Line 22, “no water added”

Line 43, this first sentence is too general. Because the first paragraph is describing loss of arthropods and the second paragraph is about bird species. I would suggest to make this first sentence more specific like “Biodiversity in insects…”

Line 56, vague, positive effects on what aspect for arthropods and birds? Mortality? Population?

Line 58, “which represents…”;and this sentence needs to be more detailed. Abundance of species being good indicators or other factors about bird population?

Line 61, similar pattern? The same declining rate? Better change to “Same declines in….” or listed the range of decline rate for those species.

Line 66-69, add “…improving food availability for bird”

Line 69, “the rapid rate of declines in bird population…”

Line 70, use “immediately” instead of “short-term”? and how authors define the short-term vs. long-term? within a year (seasonal) vs. more than 10 years?

Line 72, this first sentence is not a good topic sentence for this paragraph. “we hypothesized that..” usually follows the sentences of objectives. This paragraph should be a short summary of published papers concluding wetter soils have larger arthropods biomass.

Line 82-88, this paragraph should be put in the Method section, not here

Line 89-92, authors should have a better paragraph describing the objectives and hypothesis. Again, put hypothesis next to the objectives.

Figure 1B, increase the symbol sizes to be readable.

Table 4. two digits for p value should be enough

Experimental design

Line 95-102, should this paragraph come next to the paragraph of “study site”? and authors should provide the background of why they need this analysis in the Introduction. How this result helps interpret this study? It is very sudden to see this paragraph at the Method section.

In the paragraph of “study site”, it is better to describe the climate (annual temperature, precipitation) because soil moisture is related to climate.

Line 143-146, what was the measurement interval for soil moisture? Every hour?

Line 167, was there an equation built for each species?

Line 188-189, why these three classes are not continuous? So there will be no arthropods at the length of 3.5 mm or 1.5 mm or less than 1 mm?

Validity of the findings

Overall, in the Result section, I don't think authors need to show all ANOVA, F value, R2, only provided the p value should be satisfying.

Line 244, missing the magnitude in the near-water treatment?

Line 247, omit this sentence, put p value next to the last sentence

Line 249, “…was observed in the control...”

Line 251-253, this correlation test was not mentioned in the "data analysis" in the Method section

Line 279,285, p values here?

Line 321-326 and line 373-383, those two paragraphs should be merged together because they both discussed the landscape analysis

Line 356-358, this is an interesting point, is there any published paper supporting this point?

Line 391, this discussion point can be merged with line 367? as this site might not be ideal for studying arthropods because the majority of them were in small sizes and were not suitable for energy intake for birds. Thus, repeating the experiment in a new site with bigger sized arthropods would be ideal.

Line 397-399 is the same sentence as line 72-74, Please use different sentences.

Line 404, change to "...was negligible, probably because the arthropods..."?

Additional comments

This paper reports the effects of irrigation and near-water treatment on arthropods species and biomass, soil temperature, moisture and resistance at a farm in the Netherlands from May to June. This paper found an improvement in soil conditions under occasional irrigation but not in arthropods abundance probably because of their small sizes. This paper adds some value to the society and should be considered accepted with the above questions addressed.

---

## Round 0.2 · Major Revisions

Dear Dr. De Felici and colleagues:

Thanks for submitting your manuscript to PeerJ. I have now received three independent reviews of your work, and as you will see, two reviewers are satisfied with your revision, but one reviewer (R2) remains concerned about the experimental design.

Please read over the comments by reviewer 2 and respond to me as soon as possible. If you can address this by modifying the manuscript, please do so and upload your revised manuscript. I will not re-send for review but will make my decision after assessing your response.

Best wishes,

-joe

Reviewer 1 ·

Basic reporting

I am satisfied with the changes made by the author. I recommend accepting the manuscript.

Experimental design

ok

Validity of the findings

ok

Reviewer 2 ·

Basic reporting

The basic reporting of the manuscript is fine. Previous concerns about details of the writing appear to have been addressed.

Experimental design

I continue to believe that the experimental design is inadequate; authors have not convinced me otherwise. I appreciate the limitations that authors address at the beginning of the Discussion, but this hedge wording does not make up for a poorly replicated design or sampling arthropod for just a few months after wetting. Insect life cycles in northern Europe are often univoltine, so sampling ignored large portions of he community. Moreover, citing a published study that also has an inadequate design does not justify also using an inadequate design.

Validity of the findings

Results continue to be questionable given the poor design.

Reviewer 3 ·

Basic reporting

None

Experimental design

None

Validity of the findings

None

Additional comments

This paper has been well revised based on reviewers' comments. I really appreciated authors' hardworking. I don't have any further comments in regards of this revised version.

---

## Round 0.3 · Minor Revisions

Dear Dr. De Felici and colleagues:

Thanks for revising your manuscript. Before moving towards acceptance, I would like you to provide a substantial statement about the low (spatial and temporal) replication and the fact that, due to this, your work is a very preliminary study.

Your study is a good start, allowing your group to conduct a larger-scale experiment (perhaps five replicates in each field). This work is not overly expensive in terms of equipment or analysis – mostly in terms of time. You could propose a “future directions” path whereby the experiments are conducted over the course of two or three years.

Taking this course of action with your manuscript will help ease potential concerns by some readers similar to those of reviewer 2.

Thanks, and I look forward to seeing your slightly modified revision. I believe your work will be accepted for publication in PeerJ once this issue is addressed.

Best,

-joe

Reviewer 1 ·

Basic reporting

The abstract is too long it should be concise.
Although the info in the discussion section is sufficient and it convinces the aim of the study, but the presentation is not in a proper flow, especially the starting of the discussion section.
In conclusion section at line, no 434 author starts a sentence with the phrase "We hypothesized that ....." How come the author is hypothesized in spite of concluding in the conclusion. My suggestion to the author is to read the conclusion of a few published literature and modify the conclusion section.
English also needs brushing.

Experimental design

Experiment is well designed

Validity of the findings

'no comment'

Additional comments

The abstract is too long it should be concise.
Although the info in the discussion section is sufficient and it convinces the aim of the study, but the presentation is not in a proper flow, especially the starting of the discussion section.
In conclusion section at line, no 434 author starts a sentence with the phrase "We hypothesized that ....." How come the author is hypothesized in spite of concluding in the conclusion. My suggestion to the author is to read the conclusion of a few published literature and modify the conclusion section.
English also needs brushing.

Reviewer 3 ·

Basic reporting

None

Experimental design

None

Validity of the findings

None

Additional comments

None

---

## Round 0.4 · accepted · Accept

Dear Dr. De Felici and colleagues:

Thanks for revising your manuscript to PeerJ, and for addressing the concerns raised by the reviewers. I now believe that your manuscript is suitable for publication. Congratulations! I look forward to seeing this work in print, and I anticipate it being an important resource for research communities studying arthropod diversity in response to short- and long-term soil wetting in dairy grasslands.

Thanks again for choosing PeerJ to publish such important work.

-joe